# A Bayesian Perspective on Training Speed and Model Selection

**Clare Lyle** [†]           **Lisa Schut**[†]

**Binxin Ru**[†]       **Yarin Gal**[†]       **Mark van der Wilk**[‡]

## Abstract

We take a Bayesian perspective to illustrate a connection between training speed and the marginal likelihood in linear models. This provides two major insights: first, that a measure of a model's training speed can be used to estimate its marginal likelihood. Second, that this measure, under certain conditions, predicts the relative weighting of models in linear model combinations trained to minimize a regression loss. We verify our results in model selection tasks for linear models and for the infinite-width limit of deep neural networks. We further provide encouraging empirical evidence that the intuition developed in these settings also holds for deep neural networks trained with stochastic gradient descent. Our results suggest a promising new direction towards explaining why neural networks trained with stochastic gradient descent are biased towards functions that generalize well.

## 1  Introduction

Choosing the right inductive bias for a machine learning model, such as convolutional structure for an image dataset, is critical for good generalization. The problem of *model selection* concerns itself with identifying good inductive biases for a given dataset. In Bayesian inference, the marginal likelihood (ML) provides a principled tool for model selection. In contrast to cross-validation, for which computing gradients is cumbersome, the ML can be conveniently maximised using gradients when its computation is tractable. Unfortunately, computing the marginal likelihood for complex models such as neural networks is typically *in*tractable. Workarounds such as variational inference suffer from expensive optimization of many parameters in the variational distribution and differ significantly from standard training methods for Deep Neural Networks (DNNs), which optimize a single parameter sample from initialization. A method for estimating the ML that closely follows standard optimization schemes would pave the way for new practical model selection procedures, yet remains an open problem.

A separate line of work aims to perform model selection by predicting a model's test set performance. This has led to theoretical and empirical results connecting training speed and generalization error [17, 21]. This connection has yet to be fully explained, as most generalization bounds in the literature depend only on the final weights obtained by optimization, rather than on the trajectory taken during training, and therefore are unable to capture this relationship. Understanding the link between training speed, optimization and generalization thus presents a promising step towards developing a theory of generalization which can explain the empirical performance of neural networks.

In this work, we show that the above two lines of inquiry are in fact deeply connected. We investigate the connection between the log ML and the sum of predictive log likelihoods of datapoints, conditioned on preceding data in the dataset. This perspective reveals a family of estimators of the log

---

[†]OATML Group, University of Oxford. Correspondence to `clare.lyle@cs.ox.ac.uk`
[‡]Imperial College London

ML which depend only on predictions sampled from the posterior of an iterative Bayesian updating procedure. We study the proposed estimator family in the context of linear models, where we can conclusively analyze its theoretical properties. Leveraging the fact that gradient descent can produce exact posterior samples for linear models [31] and the infinite-width limit of deep neural networks [7, 26], we show that this estimator can be viewed as the sum of a subset of the model's training losses in an iterative optimization procedure. This immediately yields an interpretation of marginal likelihood estimation as measuring a notion of training speed in linear models. We further show that this notion of training speed is predictive of the weight assigned to a model in a linear model combination trained with gradient descent, hinting at a potential explanation for the bias of gradient descent towards models that generalize well in more complex settings.

We demonstrate the utility of the estimator through empirical evaluations on a range of model selection problems, confirming that it can effectively approximate the marginal likelihood of a model. Finally, we empirically evaluate whether our theoretical results for linear models may have explanatory power for more complex models. We find that an analogue of our estimator for DNNs trained with stochastic gradient descent is predictive of both final test accuracy and the final weight assigned to the model after training a linear model combination. Our findings in the deep learning setting hint at a promising avenue of future work in explaining the empirical generalization performance of DNNs.

## 2  Background and Related Work

### 2.1  Bayesian Parameter Inference

A Bayesian model $\mathcal{M}$ is defined by a prior distribution over parameters $\theta$, $P(\theta|\mathcal{M})$, and a prediction map from parameters $\theta$ to a likelihood over the data $\mathcal{D}$, $P(\mathcal{D}|\theta, \mathcal{M})$. Parameter fitting in the Bayesian framework entails finding the posterior distribution $P(\theta|\mathcal{D})$, which yields robust and principled uncertainty estimates. Though exact inference is possible for certain models like Gaussian processes (GPs) [38], it is intractable for DNNs. Here approximations such as variational inference [4] are used [14, 5, 27, 16, 9], to improve robustness and obtain useful uncertainty estimates.

Variational approximations require optimisation over the parameters of the approximate posterior distribution. This optimization over distributions changes the loss landscape, and is significantly slower than the pointwise optimization used in standard DNNs. Pointwise optimization methods inspired by Bayesian posterior sampling can produce similar variation and uncertainty estimates as variational inference, while improving computational efficiency [45, 30, 29]. An appealing example of this is ensembling [25], which works by training a collection models in the usual pointwise manner, starting from $k$ independently initialized points.

In the case of linear models, this is exactly equivalent to Bayesian inference, as this sample-then-optimize approach yields exact posterior samples [31, 36]. He et al. [18] extend this approach to obtain posterior samples from DNNs in the infinite-width limit.

### 2.2  Bayesian Model Selection

In addition to finding model parameters, Bayesian inference can also perform *model selection* over different inductive biases, which are specified through both model structure (e.g. convolutional vs fully connected) and the prior distribution on parameters. The Bayesian approach relies on finding the posterior over models $P(\mathcal{M}|\mathcal{D})$, which uses the *marginal likelihood* (ML) as its likelihood function:

$$P(\mathcal{D}|\mathcal{M}) = \int_\theta P(\mathcal{D}|\theta)P(\theta|\mathcal{M}_i)d\theta = \mathbb{E}_{P(\theta|\mathcal{M})}P(\mathcal{D}|\theta). \tag{1}$$

Instead of computing the full posterior, it is common to select the model with the highest marginal likelihood. This is known as type-II maximum likelihood [27, 28] and is less prone to overfitting than performing maximum likelihood over the parameters and model combined. This is because the marginal likelihood is able to trade off between model fit and model complexity [39]. Maximising the ML is standard procedure when it is easy to compute. For example, in Gaussian processes it used to set simple model parameters like smoothness [38], while recent work has demonstrated that complex inductive biases in the form of invariances can also be learned [44].

For many deep models, computing Equation 1 is intractable, and obtaining approximations that are accurate enough for model selection and that scale to complex models is an active area of research [23]. In general, variational lower bounds that scale are too loose when applied to DNNs [5]. Deep

Gaussian processes provide a case where the bounds do work [6, 8], but heavy computational load holds performance several years behind deep learning. While ensembling methods provide useful uncertainty estimates and improve the computational efficiency of the variational approach, they have not yet provided a solution for Bayesian model selection.

## 2.3 Generalization and Risk Minimization

Bayesian model selection addresses a subtly different problem from the risk minimization framework used in many learning problems. Nonetheless, the two are closely related; Germain et al. [15] show that in some cases optimizing a PAC-Bayesian risk bound is equivalent to maximizing the marginal likelihood of a Bayesian model. In practice, maximizing an approximation of the marginal likelihood in DNNs trained with SGD can improve generalization performance [41]. More recently, Arora et al. [1] computed a data-dependent complexity measure which resembles the data-fit term in the marginal likelihood of a Bayesian model and which relates to optimization speed, hinting at a potential connection between the two.

At the same time, generalization in deep neural networks (DNNs) remains mysterious, with classical learning-theoretic bounds failing to predict the impressive generalization performance of DNNs [47, 33]. Recent work has shown that DNNs are biased towards functions that are 'simple', for various definitions of simplicity [22, 13, 43, 42]. PAC-Bayesian generalization bounds, which can quantify a broad range of definitions of complexity, can attain non-vacuous values [32, 10, 11], but nonetheless exhibit only modest correlation with generalization error [21]. These bounds depend only on the final distribution over parameters after training; promising alternatives consider properties of the trajectory taken by a model during optimization [17, 35]. This trajectory-based perspective is a promising step towards explaining the correlation between the number of training steps required for a model to minimize its objective function and its final generalization performance observed in a broad range of empirical analyses [21, 3, 34, 40].

## 3 Marginal Likelihood Estimation with Training Statistics

In this section, we investigate the equivalence between the marginal likelihood (ML) and a notion of training speed in models trained with an exact Bayesian updating procedure. For linear models and infinitely wide neural networks, exact Bayesian updating can be done using gradient descent optimisation. For these cases, we derive an estimator of the marginal likelihood which **1)** is related to how quickly a model learns from data, **2)** only depends on statistics that can be measured during pointwise gradient-based parameter estimation, and **3)** becomes tighter for ensembles consisting of multiple parameter samples. We also investigate how gradient-based optimization of a linear model combination can implicitly perform approximate Bayesian model selection in Section 3.3.

### 3.1 Training Speed and the Marginal Likelihood

Let $\mathcal{D}$ denote a dataset of the form $\mathcal{D} = (\mathcal{D}_i)_{i=1}^n = (x_i, y_i)_{i=1}^n$, and let $\mathcal{D}_{<i} = (\mathcal{D}_j)_{j=1}^{i-1}$ with $\mathcal{D}_{<1} = \emptyset$. We will abbreviate $P(\mathcal{D}|\mathcal{M}) := P(\mathcal{D})$ when considering a single model $\mathcal{M}$. Observe that $P(\mathcal{D}) = \prod_{i=1}^n P(\mathcal{D}_i|\mathcal{D}_{<i})$ to get the following form of the *log* marginal likelihood:

$$\log P(\mathcal{D}) = \log \prod_{i=1}^n P(\mathcal{D}_i|\mathcal{D}_{<i}) = \sum_{i=1}^n \log P(\mathcal{D}_i|\mathcal{D}_{<i}) = \sum_{i=1}^n \log[\mathbb{E}_{P(\theta|\mathcal{D}_{<i})}P(\mathcal{D}_i|\theta)]. \quad (2)$$

If we define training speed as the number of data points required by a model to form an accurate posterior, then models which train faster – i.e. whose posteriors assign high likelihood to the data after conditioning on only a few data points – will obtain a higher marginal likelihood. Interpreting the negative log posterior predictive probability $\log P(\mathcal{D}_i|\mathcal{D}_{<i})$ of each data point as a loss function, the log ML then takes the form of the sum over the losses incurred by each data point during training, i.e. the area under a training curve defined by a Bayesian updating procedure.

### 3.2 Unbiased Estimation of a Lower Bound

In practice, computing $\log P(\mathcal{D}_i|\mathcal{D}_{<i})$ may be intractable, necessitating approximate methods to estimate the model evidence. In our analysis, we are interested in estimators of $\log P(\mathcal{D})$ computed

by drawing $k$ samples of $\theta \sim P(\theta|\mathcal{D}_{<i})$ for each $i = 1, \ldots, n$. We can directly estimate a lower bound $\mathcal{L}(\mathcal{D}) = \sum_{i=1}^{n} \mathbb{E}[\log P(\mathcal{D}_i|\mathcal{D}_{<i})]$ using the log likelihoods of these samples

$$\hat{\mathcal{L}}(\mathcal{D}) = \sum_{i=1}^{n} \frac{1}{k} \sum_{j=1}^{k} \log P(\mathcal{D}_i|\theta_j^i). \tag{3}$$

This will produce a biased estimate of the log marginal likelihood due to Jensen's inequality. We can get a tighter lower bound by first estimating $\mathbb{E}[\log P(\mathcal{D}_i|\theta)]$ using our posterior samples before applying the logarithm, obtaining

$$\hat{\mathcal{L}}_k(\mathcal{D}) = \sum_{i=1}^{n} \log \frac{1}{k} \sum_{j=1}^{k} P(\mathcal{D}_i|\theta_j^i). \tag{4}$$

**Proposition 3.1.** Both $\hat{\mathcal{L}}$ and $\hat{\mathcal{L}}_k$ as defined in Equation 4 are estimators of lower bounds on the log marginal likelihood; that is

$$\mathbb{E}[\hat{\mathcal{L}}(\mathcal{D})] = \mathcal{L}(\mathcal{D}) \le \log P(\mathcal{D}) \quad \text{and} \quad \mathbb{E}[\hat{\mathcal{L}}_k(\mathcal{D})] = \mathcal{L}_k(\mathcal{D}) \le \log P(\mathcal{D}) . \tag{5}$$

Further, the bias term in $\mathcal{L}$ can be quantified as follows.

$$\mathcal{L}(\mathcal{D}) = \log P(\mathcal{D}) - \sum_{i=1}^{n} \mathrm{KL}(P(\theta|\mathcal{D}_{<i})||P(\theta|\mathcal{D}_{<i+1})) \tag{6}$$

We include the proof of this and future results in Appendix A. We observe that both lower bound estimators exhibit decreased variance when using multiple posterior samples; however, $\hat{\mathcal{L}}_k$ also exhibits decreasing bias (with respect to the log ML) as $k$ increases; each $k$ defines a distinct lower bound $\mathcal{L}_k = \mathbb{E}[\hat{\mathcal{L}}_k]$ on $\log P(\mathcal{D})$. The gap induced by the lower bound $\mathcal{L}(\mathcal{D})$ is characterized by the information gain each data point provides to the model about the posterior, as given by the Kullback-Leibler (KL) divergence [24] between the posterior at time $i$ and the posterior at time $i + 1$. Thus, while $\mathcal{L}$ has a Bayesian interpretation it is arguably more closely aligned with the minimum description length notion of model complexity [19].

When the posterior predictive distribution of our model is Gaussian, we consider a third approach which, unlike the previous two methods, also applies to noiseless models. Let $\mathcal{D} = (X_i, Y_i)_{i=1}^{n}$, and $(\theta_j^i)_{j=1}^{k}$ be $k$ parameter samples from $P(\theta|\mathcal{D}_{<i})$. We assume a mapping $f : \Theta \times X \to Y$ such that sampling parameters $\theta$ and computing $f(\theta, X_i)$ is equivalent to sampling from the posterior $P(\cdot|\mathcal{D}_{<i}, X_i)$. We can then obtain the following estimator of a lower bound on $\log \mathcal{P}(\mathcal{D})$.

**Proposition 3.2.** Let $P(Y_i|\mathcal{D}_{<i}, X_i) = \mathcal{N}(\mu_i, \sigma_i^2)$ for some $\mu_i, \sigma_i^2$. Define the standard mean and variance estimators $\hat{\mu}_i = \frac{1}{N} \sum_{j=1}^{N} f(\theta_j^i, x_i)$ and $\hat{\sigma}_i^2 = \frac{1}{N-1} \sum (f(\theta_j^i, x_i) - \hat{\mu})^2$. Then the estimator

$$\hat{\mathcal{L}}_S(\mathcal{D}) = \sum_{i=1}^{n} \log P(Y_i|\hat{\mu}_i, \hat{\sigma}_i^2) \tag{7}$$

is a lower bound on the log ML: i.e. $\mathbb{E}[\hat{\mathcal{L}}_S(\mathcal{D})] \le \log P(\mathcal{D})$.

We provide an empirical evaluation of the rankings provided by the different estimators in Section 4. We find that $\hat{\mathcal{L}}_S$ exhibits the least bias in the presence of limited samples from the posterior, though we emphasize its limitation to Gaussian posteriors; for more general posterior distributions, $\hat{\mathcal{L}}_k$ minimizes bias while still estimating a lower bound.

### 3.2.1 Lower bounds via gradient descent trajectories

The bounds on the marginal likelihood we introduced in the previous section required samples from the sequence of posteriors as data points were incrementally added $p(\theta|\mathcal{D}_{<i})$. Ensembles of linear models trained with gradient descent yield samples from the model posterior. We now show that we can use these samples to estimate the log ML using the estimators introduced in the previous section.

We will consider the Bayesian linear regression problem of modelling data $\mathcal{D} = (X_i, Y_i)_{i=1}^{n}$ assumed to be generated by the process $Y = \theta^\top \Phi(X) + \epsilon \sim \mathcal{N}(0, \sigma_N^2 I)$ for some unknown $\theta$, known $\sigma_N^2$,

and feature map $\Phi$. Typically, a Gaussian prior is placed on $\theta$; this prior is then updated as data points are seen to obtain a posterior over parameters. In the overparmeterised, noiseless linear regression setting, Matthews et al. [31] show that the distribution over parameters $\theta$ obtained by sampling from the prior on $\theta_0$ and running gradient descent to convergence on the data $\mathcal{D}_{<i}$ is equivalent to sampling from the posterior conditioned on $\mathcal{D}_{<i}$. Osband et al. [36] extend this result to posteriors which include observation noise $\sigma_N^2 \neq 0$ under the assumption that the targets $Y_i$ are themselves noiseless observations.

---

**Algorithm 1:** Marginal Likelihood Estimation for Linear Models

---

**Input:** A dataset $\mathcal{D} = (x_i, y_i)_{i=1}^n$, parameters $\mu_0, \sigma_0^2, \sigma_N^2$
**Result:** An estimate of $\mathcal{L}(\mathcal{D})$
$\theta_t \leftarrow \theta_0 \sim \mathcal{N}(\mu_0, \sigma_0^2); \quad \tilde{Y} \leftarrow Y + \epsilon \sim \mathcal{N}(0, \sigma_N^2); \quad \text{sumLoss} \leftarrow 0$ ;
$\ell(\mathcal{D}_{\leq i}, w) \leftarrow \|\tilde{Y}_{\leq i} - \theta^\top X_{\leq i}\|_2^2 + \frac{\sigma_N^2}{\theta_0^2}\|\theta - \theta_0\|_2^2$;
**for** $\mathcal{D}_i \in \mathcal{D}$ **do**
> $\text{sumLoss} = \text{sumLoss} + \frac{(\theta_t^\top x_i - y_i)^2}{2\sigma_N^2}$ ;
> $\theta_t \leftarrow \text{GradientDescent}(\ell, \theta_t, \mathcal{D}_{\leq i})$ ;

**end**
**return** sumLoss

---

We can use this procedure to obtain posterior samples for our estimators by iteratively running sample-then-optimize on the sets $\mathcal{D}_{<i}$. Algorithm 1 outlines our approach, which uses sample-then-optimize on iterative subsets of the data to obtain the necessary posterior samples for our estimator. Theorem 3.3 shows that this procedure yields an unbiased estimate of $\mathcal{L}(\mathcal{D})$ when a single prior sample is used, and an unbiased estimate of $\hat{\mathcal{L}}_k(\mathcal{D})$ when an ensemble of $k$ models are trained in parallel.

**Theorem 3.3.** Let $\mathcal{D} = (X_i, Y_i)_{i=1}^n$ and let $(\theta_j^i)_{i,j=1}^{n,J}$ be generated by the procedure outlined above. Then the estimators $\hat{\mathcal{L}}, \hat{\mathcal{L}}_S$, and $\hat{\mathcal{L}}_k$, applied to the collection $(\theta_j^i)$, are lower bounds on $\log P(\mathcal{D})$. Further, expressing $-\log P(\mathcal{D}_i|\theta)$ as the $\ell_2$ regression loss plus a constant, we then obtain

$$\log P(\mathcal{D}) \geq \sum_{i=1}^n \mathbb{E}_{\theta_i \sim P(\cdot|\mathcal{D}_{<i})}[\log P(\mathcal{D}_i|\theta_i)] = \mathbb{E}\sum_{i=1}^n -\ell_2(\mathcal{D}_i, \theta_i) + c = \mathcal{L}(\mathcal{D}) \qquad (8)$$

We highlight that Theorem 3.3 precisely characterizes the lower bound on the marginal likelihood as a sum of 'training losses' based on the regression loss $\ell_2(\mathcal{D}_i, \theta_i)$.

### 3.2.2 From Linear Models to Infinite Neural Networks

Beyond linear models, our estimators can further perform model selection in the infinite-width limit of neural networks. Using the optimization procedure described by He et al. [18], we can obtain an exact posterior sample from a GP given by the neural tangent kernel [20]. The iterative training procedure described in Algorithm 1 will thus yield a lower bound on the marginal likelihood of this GP using sampled losses from the optimization trajectory of the neural network. We evaluate this bound in Section 4, and formalize this argument in the following corollary.

**Corollary 3.4.** Let $\mathcal{D}$ be a dataset indexed by our standard notation. Let $f_0$ be sampled from an infinitely wide neural network architecture $\mathcal{F}$ under some initialization distribution, and let $f_\infty^i$ be the limiting solution under the training dynamics defined by He et al. [18] applied to the initialization $f_0$ and using data $\mathcal{D}_{<i}$. Let $K_\infty$ denote the neural tangent kernel for $\mathcal{F}$, and $\mathcal{M} = GP(0, K_\infty)$ the induced Gaussian Process. Then $f_\infty^i \sim P(f|\mathcal{D}_{<i}, \mathcal{M})$, and in the limit of infinite training time, the iterative sample-then-optimize procedure yields an unbiased estimate of $\mathcal{L}(\mathcal{D}|\mathcal{M})$. Letting $\ell_2$ denote the scaled squared $\ell_2$ regression loss and $c$ be a constant, we obtain as a direct corollary of Theorem 3.3

$$P(\mathcal{D}) \geq \mathbb{E}_{f_\infty^i \sim P(\cdot|\mathcal{D}_{<i})}[\log P(\mathcal{D}_i|\theta_i)] = \mathbb{E}\sum_{i=1}^n -\ell_2(\mathcal{D}_i, f_i) + c = \mathcal{L}(\mathcal{D}) . \qquad (9)$$

This result provides an additional view on the link between training speed and generalisation in wide neural networks noted by Arora et al. [1], who analysed the convergence of gradient descent. They

compute a PAC generalization bound which a features the data complexity term equal to that in the marginal likelihood of a Gaussian process Rasmussen [38]. This term provides a bound on the rate of convergence of gradient descent, whereas our notion of training speed is more closely related to sample complexity and makes the connection to the marginal likelihood more explicit.

It is natural to ask if such a Bayesian interpretation of the sum over training losses can be extended to non-linear models trained with stochastic gradient descent. Although SGD lacks the exact posterior sampling interpretation of our algorithm, we conjecture a similar underlying mechanism connecting the sum over training losses and generalization. Just as the marginal likelihood measures how well model updates based on previous data points generalize to a new unseen data point, the sum of training losses measures how well parameter updates based on one mini-batch generalize to the rest of the training data. If the update generalizes well, we expect to see a sharper decrease in the training loss, i.e. for the model to train more quickly and exhibit a lower sum over training losses. This intuition can be related to the notion of 'stiffness' proposed by Fort et al. [12]. We provide empirical evidence supporting our hypothesis in Section 4.2.

### 3.3   Bayesian Model Selection and Optimization

The estimator $\mathcal{L}(\mathcal{D})$ reveals an intriguing connection between pruning in linear model combinations and Bayesian model selection. We assume a data set $\mathcal{D} = (X_i, Y_i)_{i=1}^n$ and a collection of $k$ models $\mathcal{M}_1, \ldots, \mathcal{M}_k$. A linear regressor $w$ is trained to fit the posterior predictive distributions of the models to the target $Y_i$; i.e. to regress on the dataset

$$(\Phi, Y) = \left( \phi_i = (\hat{Y}_1^i, \ldots, \hat{Y}_n^i), Y_i \right)_{i=1}^n \text{ with } \hat{Y}_j^i \sim P(\hat{Y}|\mathcal{D}_{<i}, X_i, \mathcal{M}_j). \tag{10}$$

The following result shows that the optimal linear regressor on this data generating distribution assigns the highest weight to the model with the highest $\mathcal{L}(\mathcal{D})$ whenever the model errors are independent. This shows that magnitude pruning in a linear model combination is equivalent to approximate Bayesian model selection, under certain assumptions on the models.

**Proposition 3.5.** Let $\mathcal{M}_1, \ldots, \mathcal{M}_k$ be Bayesian linear regression models with fixed noise variance $\sigma_N^2$ and Gaussian likelihoods. Let $\Phi$ be a (random) matrix of posterior prediction samples, of the form $\Phi[i, j] = \hat{y}_i^j \sim P(y_j|\mathcal{D}_{<j}, x_j, \mathcal{M}_i)$. Suppose the following two conditions on the columns of $\Phi$ are satisfied: $\mathbb{E}\langle\Phi[:, i], y\rangle = \mathbb{E}\langle\Phi[:, j], y\rangle$ for all $i, j$, and $\mathbb{E}\langle\Pi_{y^\perp}\phi_i, \Pi_{y^\perp}\phi_j\rangle = 0$. Let $w^*$ denote the least-squares solution to the regression problem $\min_w \mathbb{E}_\Phi\|\Phi w - y\|^2$. Then the following holds

$$\arg\max_i w_i^* = \arg\max_i \mathcal{L}(\mathcal{D}|\mathcal{M}_i) \qquad \forall w^* = \arg\min_w \mathbb{E}\|\Phi w - y\|^2 . \tag{11}$$

The assumption on the independence of model errors is crucial in the proof of this result: families of models with large and complementary systematic biases may not exhibit this behaviour. We observe in Section 4 that the conditions of Proposition 1 are approximately satisfied in a variety of model comparison problems, and running SGD on a linear combination of Bayesian models still leads to solutions that approximate Bayesian model selection. We conjecture that analogous phenomena occur during training within a neural network. The proof of Proposition 3.5 depends on the observation that, given a collection of features, the best least-squares predictor will assign the greatest weight to the feature that best predicts the training data. While neural networks are not linear ensembles of fixed models, we conjecture that, especially for later layers of the network, a similar phenomenon will occur wherein weights from nodes that are more predictive of the target values over the course of training will be assigned higher magnitudes. We empirically investigate this hypothesis in Section 4.2.

## 4   Empirical Evaluation

Section 3 focused on two key ideas: that training statistics can be used as an estimator for a Bayesian model's marginal likelihood (or a lower bound thereof), and that gradient descent on a linear ensemble implicitly arrives at the same ranking as this estimator in the infinite-sample, infinite-training-time limit. We further conjectured that similar phenomena may also hold for deep neural networks. We now illustrate these ideas in a range of settings. Section 4.1 provides confirmation and quantification of our results for linear models, the model class for which we have theoretical guarantees, while Section 4.2 provides preliminary empirical confirmation that the mechanisms at work in linear models also appear in DNNs.

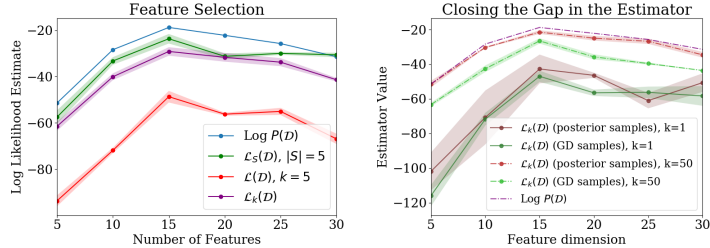

Figure 1: Left: ranking according to $\log P(\mathcal{D})$, $\mathcal{L}(\mathcal{D})$ with exact posterior samples, and $\mathcal{L}(\mathcal{D})$ computed on samples generated by gradient descent. Right: gap between true marginal likelihood and $\mathcal{L}_k(\mathcal{D})$ estimator shrinks as a function of $k$ for both exact and gradient descent-generated samples.

## 4.1 Bayesian Model Selection

While we have shown that our estimators correspond to lower bounds on the marginal likelihood, we would also like the relative rankings of models given by our estimator to correlate with those assigned by the marginal likelihood. We evaluate this correlation in a variety of linear model selection problems. We consider three model selection problems; for space we focus on one, feature dimension selection, and provide full details and evaluations on the other two tasks in Appendix B.1.

For the feature dimension selection task, we construct a synthetic dataset inspired by Wilson and Izmailov [46] of the form $(\mathbf{X}, \mathbf{y})$, where $x_i = (y_i + \epsilon_1, y_i + \ldots, y_i + \epsilon_{15}, \epsilon_{16}, \ldots, \epsilon_{30})$, and consider a set of models $\{\mathcal{M}_k\}$ with feature embeddings $\phi_k(x_i) = x_i[1, \ldots, k]$. The optimal model in this setting is the one which uses exactly the set of 'informative' features $x[1, \ldots, 15]$.

We first evaluate the relative rankings given by the true marginal likelihood with those given by our estimators. We compare $\mathcal{L}_S$, $\mathcal{L}$ and $\mathcal{L}_k$; we first observe that all methods agree on the optimal model: this is a consistent finding across all of the model selection tasks we considered. While all methods lower bound the log marginal likelihood, $\mathcal{L}_k(\mathcal{D})$ and $\mathcal{L}_S(\mathcal{D})$ exhibit a reduced gap compared to the naive lower bound. In the rightmost plot of Figure 1, we further quantify the reduction in the bias of the estimator $\mathcal{L}_k(\mathcal{D})$ described in Section 3. We use exact posterior samples (which we denote in the figure simply as posterior samples) and approximate posterior samples generated by the gradient descent procedure outlined in Algorithm 1 using a fixed step size and thus inducing some approximation error. We find that both sampling procedures exhibit decreasing bias as the number of samples $k$ is increased, with the exact sampling procedure exhibiting a slightly smaller gap than the approximate sampling procedure.

We next empirically evaluate the claims of Proposition 3.5 in settings with relaxed assumptions. We compare the ranking given by the true log marginal likelihood, the estimated $\mathcal{L}(\mathcal{D})$, and the weight assigned to each model by the trained linear regressor. We consider three variations on how sampled predictions from each model are drawn to generate the features $\phi_i$: sampling the prediction for point $\hat{Y}_i$ from $P(\hat{Y}_i | \mathcal{D}_{<i})$ ('concurrent sampling' – this is the setting of Proposition 3.5), as well as two baselines: the posterior $P(\hat{Y}_i | \mathcal{D})$ ('posterior sampling'), and the prior $P(\hat{Y}_i)$ ('prior sampling'). We find that the rankings of the marginal likelihood, its lower bound, and of the ranking given by concurrent optimization all agree on the best model in all three of the model selection problems outlined previously, while the prior and posterior sampling procedure baselines do not exhibit a consistent ranking with the log ML. We visualize these results for the feature dimension selection problem in Figure 2; full results are shown in Figure 5.

We further illustrate how the $\mathcal{L}(\mathcal{D})$ estimator can select inductive biases in the infinite-width neural network regime in Figure 2. Here we evaluate the relative change in the log ML of a Gaussian Process induced by a fully-connected MLP (MLP-NTK-GP) and a convolutional neural network (Conv-NTK-GP) which performs regression on the MNIST dataset. The fully-connected model sees a consistent decrease in its log ML with each additional data point added to the dataset, whereas the convolutional model sees the incremental change in its log ML become less negative as more data points are added as a result of its implicit bias, as well as a much higher incremental change in its log ML from the start of training. This leads to the Conv-NTK-GP having a higher value for $\mathcal{L}(\mathcal{D})$ than the MLP-NTK-GP. We provide an analogous plot evaluating $\log P(\mathcal{D})$ in the appendix.

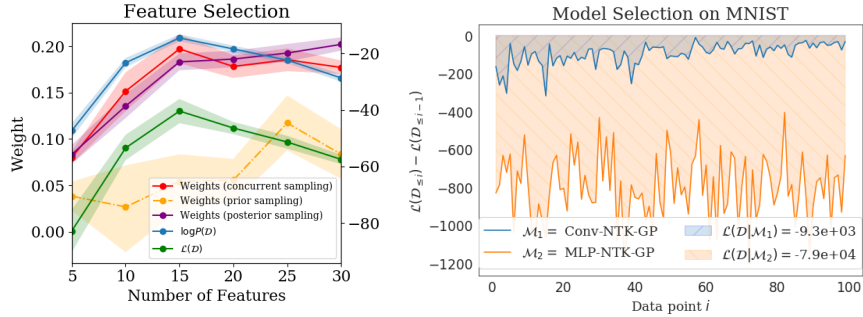

Figure 2: Left: Relative rankings given by optimize-then-prune, ML, and estimated $\mathcal{L}(\mathcal{D})$ on the feature selection problem. Right: visualizing the interpretation of $\mathcal{L}(\mathcal{D})$ as the 'area under the curve' of training losses: we plot the relative change in the estimator $\mathcal{L}(\mathcal{D}_{\leq i}) - \mathcal{L}(\mathcal{D}_{<i})$ for convolutional and fully-connected NTK-GP models, and shade their area.

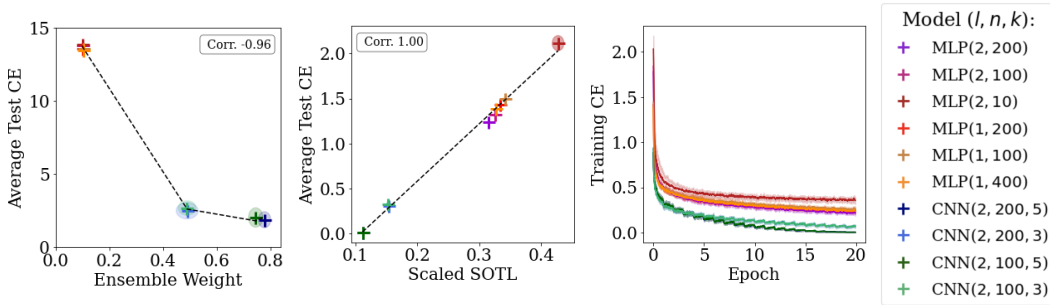

Figure 3: Linear combinations of DNNs on FashionMNIST trained. Left: ensemble weights versus the test loss for concurrent training. Middle: sum over training losses (SOTL), standardized by the number of training samples, versus test loss for parallel training. Right: training curves for the different models trained in parallel. All results are averaged over 10 runs, and standard deviations are shown by the shaded regions around each observation. The model parameters, given in the parentheses, are the number of layers ($l$), nodes per layer ($n$) and kernel size ($k$), respectively.

## 4.2 Training Speed, Ensemble Weight, and Generalization in DNNs

We now address our conjectures from Section 3, which aim to generalize our results for linear models to deep neural networks trained with SGD. Recall that our hypothesis involves translating *iterative posterior samples* to *minibatch training losses over an SGD trajectory*, and *bayesian model evidence* to *generalization error*; we conjectured that just as the sum of the log posterior likelihoods is useful for Bayesian model selection, the sum of minibatch training losses will be useful to predict generalization error. In this section, we evaluate whether this conjecture holds for a simple convolutional neural network trained on the FashionMNIST dataset. Our results provide preliminary evidence in support of this claim, and suggest that further work investigating this relationship may reveal valuable insights into how and why neural networks generalize.

### 4.2.1 Linear Combination of DNN Architectures

We first evaluate whether the sum over training losses (SOTL) obtained over an SGD trajectory correlates with a model's generalization error, and whether SOTL predicts the weight assigned to a model by a linear ensemble. To do so, we train a linear combination of DNNs with SGD to determine whether SGD upweights NNs that generalize better. Further details of the experiment can be found in Appendix B.2. Our results are summarized in Figure 3.

We observe a strong correlation between SOTL and average test cross-entropy (see Figure 3 middle column), validating that the SOTL is correlated with generalization. Further, we find that architectures with lower test error (when trained individually) are given higher weight by the linear ensembling layer – as can be seen from the left plot in Figure 3. This supports our hypothesis that *SGD favours models that generalize well*.

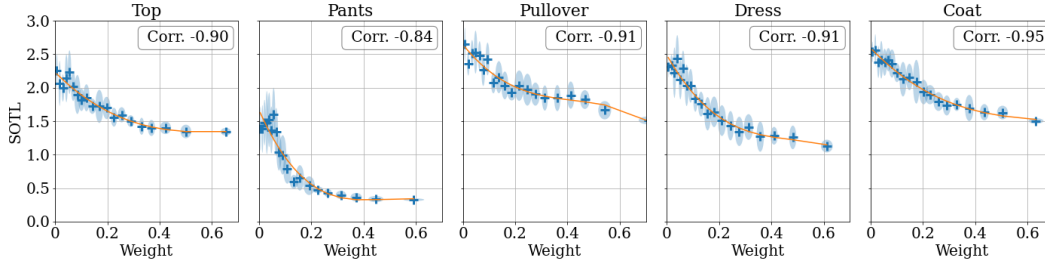

Figure 4: Weight assigned to subnetwork by SGD in a deep neural network (x-axis) versus the subnetwork performance (estimated by the sum of cross-entropy, on the y-axis) for different FashionMNIST classes. The light blue ovals denote depict $95\%$ confidence intervals, estimated over 10 seeds (i.e. $2\sigma$ for both the weight and SOTL). The orange line depicts the general trend.

### 4.2.2   Subnetwork Selection in Neural Networks

Finally, we evaluate whether our previous insights apply to submodels within a neural network, suggesting a potential mechanism which may bias SGD towards parameters with better generalization performance. Based on the previous experiments, we expect that nodes that have a lower sum over training errors (if evaluated as a classifier on their own) are favoured by gradient descent and therefore have a larger final weight than those which are less predictive of the data. If so, we can then view SGD followed by pruning (in the final linear layer of the network) as performing an approximation of a Bayesian model selection procedure. We replicate the model selection problem of the previous setting, but replace the individual models with the activations of the penultimate layer of a neural network, and replace the linear ensemble with the final linear layer of the network. Full details on the experimental set-up can be found in Appendix B.3. We find that our hypotheses hold here: SGD assigns larger weights to subnetworks that perform well, as can be seen in Figure 4. This suggests that SGD is biased towards functions that generalize well, even within a network. We find the same trend holds for CIFAR-10, which is shown in Appendix B.3.

## 5   Conclusion

In this paper, we have proposed a family of estimators of the marginal likelihood which illustrate the connection between training speed and Bayesian model selection. Because gradient descent can produce exact posterior samples in linear models, our result shows that Bayesian model selection can be done by training a linear model with gradient descent and tracking how quickly it learns. This approach also applies to the infinite-width limit of deep neural networks, whose dynamics resemble those of linear models. We further highlight a connection between magnitude-based pruning and model selection, showing that models for which our lower bound is high will be assigned more weight by an optimal linear model combination. This raises the question of whether similar mechanisms exist in finitely wide neural networks, which do not behave as linear models. We provide preliminary empirical evidence that the connections shown in linear models have predictive power towards explaining generalization and training dynamics in DNNs, suggesting a promising avenue for future work.

# 6  Broader Impact

Due to the theoretical nature of this paper, we do not foresee any immediate applications (positive or negative) that may arise from our work. However, improvement in our understanding of generalization in deep learning may lead to a host of downstream impacts which we outline briefly here for completeness, noting that the marginal effect of this paper on such broad societal and environmental impacts is likely to be very small.

1. **Safety and robustness.** Developing a stronger theoretical understanding of generalization will plausibly lead to training procedures which improve the test-set performance of deep neural networks. Improving generalization performance is crucial to ensuring that deep learning systems applied in practice behave as expected based on their training performance.

2. **Training efficiency and environmental impacts.** In principle, obtaining better estimates of model and sub-model performance could lead to more efficient training schemes, thus potentially reducing the carbon footprint of machine learning research.

3. **Bias and Fairness.** The setting of our paper, like much of the related work on generalization, does not consider out-of-distribution inputs or training under constraints. If the training dataset is biased, then a method which improves the generalization performance of the model under the i.i.d. assumption will be prone to perpetuating this bias.

## Acknowledgements

Lisa Schut was supported by the Accenture Labs and Alan Turing Institute.

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
