[Supplementary Material]

# A    Proofs of Theoretical Results

**Proposition 3.1.** Both $\hat{\mathcal{L}}$ and $\hat{\mathcal{L}}_k$ as defined in Equation 4 are estimators of lower bounds on the log marginal likelihood; that is

$$\mathbb{E}[\hat{\mathcal{L}}(\mathcal{D})] = \mathcal{L}(\mathcal{D}) \leq \log P(\mathcal{D}) \quad \text{and} \quad \mathbb{E}[\hat{\mathcal{L}}_k(\mathcal{D})] = \mathcal{L}_k(\mathcal{D}) \leq \log P(\mathcal{D}) . \tag{5}$$

Further, the bias term in $\mathcal{L}$ can be quantified as follows.

$$\mathcal{L}(\mathcal{D}) = \log P(\mathcal{D}) - \sum_{i=1}^{n} \mathrm{KL}(P(\theta|\mathcal{D}_{<i})||P(\theta|\mathcal{D}_{<i+1})) \tag{6}$$

*Proof.* The result for $\mathcal{L}$ follows from a straightforward derivation:

$$\mathcal{L}(\mathcal{D}) = \sum \int \log P(\mathcal{D}_i|\theta) dP(\theta|\mathcal{D}_{<i}) \tag{12}$$

$$= \sum \int \log[\frac{P(\mathcal{D}_i|\theta)P(\theta|\mathcal{D}_{<i})P(\mathcal{D}_i|\mathcal{D}_{<i})}{P(\theta|\mathcal{D}_{<i})P(\mathcal{D}_i|\mathcal{D}_{<i})}] dP(\theta|\mathcal{D}_{<i}) \tag{13}$$

$$= \sum \int \log \frac{P(\theta|\mathcal{D}_{\leq i}))}{P(\theta|\mathcal{D}_{<i})} dP(\theta|\mathcal{D}_{<i}) + \sum \log P(\mathcal{D}_i|\mathcal{D}_{<i}) \tag{14}$$

$$= \sum \left( \log P(\mathcal{D}_i|\mathcal{D}_{<i}) - \mathrm{KL}(P(\theta|\mathcal{D}_{<i})||P(\theta|\mathcal{D}_{\leq i})) \right) \tag{15}$$

$$= \log P(\mathcal{D}) - \sum_{i=1}^{n} \mathrm{KL}(P(\theta|\mathcal{D}_{<i})||P(\theta|\mathcal{D}_{\leq i})). \tag{16}$$

The result for $\hat{\mathcal{L}}_k$ follows immediately from Jensen's inequality, yielding

$$\sum \mathbb{E}[\log \sum_{j=1}^{k} \frac{1}{k} p(\mathcal{D}_i|\theta_j)] \leq \sum \log \mathbb{E}[\sum_{j=1}^{k} \frac{1}{k} p(\mathcal{D}_i|\theta_j)] = \sum \log \mathbb{E}[p(\mathcal{D}_i|\theta_j)] = \log P(\mathcal{D}) . \tag{17}$$

Because $\mathcal{L}_k$ applies Jensen's inequality to a random variable with decreasing variance as a function of $k$, we expect the bias of $\mathcal{L}_k$ to decrease as $k$ grows, an observation characterized in Section 4. $\square$

**Proposition 3.2.** Let $P(Y_i|\mathcal{D}_{<i}, X_i) = \mathcal{N}(\mu_i, \sigma_i^2)$ for some $\mu_i, \sigma_i^2$. Define the standard mean and variance estimators $\hat{\mu}_i = \frac{1}{N} \sum_{j=1}^{N} f(\theta_j^i, x_i)$ and $\hat{\sigma}_i^2 = \frac{1}{N-1} \sum (f(\theta_j^i, x_i) - \hat{\mu})^2$. Then the estimator

$$\hat{\mathcal{L}}_S(\mathcal{D}) = \sum_{i=1}^{n} \log P(Y_i|\hat{\mu}_i, \hat{\sigma}_i^2) \tag{7}$$

is a lower bound on the log ML: i.e. $\mathbb{E}[\hat{\mathcal{L}}_S(\mathcal{D})] \leq \log P(\mathcal{D})$.

*Proof.* To show that the sum of the estimated log likelihoods is a lower bound on the log marginal likelihood, it suffices to show that each term in the sum of the estimates is a lower bound on the corresponding term in log marginal likelihood expression. Thus, without loss of generality we consider a single data point $\mathcal{D}_i = (x, y)$ and posterior distribution $p(y|x, \mathcal{D}_{<i}) = \mathcal{N}(\mu, \sigma^2)$.

Let $y \in \mathbb{R}$, $\hat{\mu}, \hat{\sigma}$ the standard estimators for sample mean and variance given sample $\hat{Y} \in \mathbb{R}^k$ sampled from $\mathcal{N}(\mu, \sigma^2)$. We want to show

$$\mathbb{E}_{\hat{Y} \sim \mathcal{N}(\mu, \sigma^2)}[\ln p(y|\hat{\mu}, \hat{\sigma}^2)] \leq \ln p(y|\mu, \sigma^2). \tag{18}$$

We first note that $\hat{\mu}(\hat{Y}) \perp \hat{\sigma}(\hat{Y})$ for $\hat{Y}$ a collection of i.i.d. Gaussian random variables [2]. We also take advantage of the fact that the log likelihood of a Gaussian is concave with respect to its $\mu$ parameter and its $\sigma^2$ parameter. Notably, the log likelihood is *not* concave w.r.t. the joint pair $(\mu, \sigma^2)$, but because the our estimators are independent, this will not be a problem for us. We proceed as follows by first decomposing the expectation over the samples $\hat{Y}$ into an expectation over $\hat{\mu}$ and $\widehat{\sigma^2}$

$$\mathbb{E}_{X \sim \mathcal{N}(\mu, \sigma^2)}[\ln p(y|\hat{\mu}, \hat{\sigma}^2)] = \mathbb{E}_{\hat{\mu}, Y_2, \dots, Y_N} \ln p(y|\hat{\mu}, \hat{\sigma}^2) \tag{19}$$

$$= \mathbb{E}_{\hat{\mu}} \mathbb{E}_{\hat{\sigma}^2} \ln p(y|\hat{\mu}, \hat{\sigma}^2) \tag{20}$$

We apply Jensen's inequality first to the inner expectation, then to the outer.

$$\leq \mathbb{E}_{\hat{\mu}} \ln p(y|\hat{\mu}, \mathbb{E}[\hat{\sigma}^2]) = \mathbb{E}_{\hat{\mu}} \ln p(y|\hat{\mu}, \sigma^2) \qquad (21)$$

$$\leq \ln p(y|\mu, \sigma^2) \qquad (22)$$

So we obtain our lower bound. $\qquad\square$

**Theorem 3.3.** Let $\mathcal{D} = (X_i, Y_i)_{i=1}^n$ and let $(\theta_j^i)_{i,j=1}^{n,J}$ be generated by the procedure outlined above. Then the estimators $\hat{\mathcal{L}}, \hat{\mathcal{L}}_S$, and $\hat{\mathcal{L}}_k$, applied to the collection $(\theta_j^i)$, are lower bounds on $\log P(\mathcal{D})$. Further, expressing $-\log P(\mathcal{D}_i|\theta)$ as the $\ell_2$ regression loss plus a constant, we then obtain

$$\log P(\mathcal{D}) \geq \sum_{i=1}^n \mathbb{E}_{\theta_i \sim P(\cdot|\mathcal{D}_{<i})}[\log P(\mathcal{D}_i|\theta_i)] = \mathbb{E}\sum_{i=1}^n -\ell_2(\mathcal{D}_i, \theta_i) + c = \mathcal{L}(\mathcal{D}) \qquad (8)$$

*Proof.* The heavy lifting for this result has largely been achieved by Propositions 3.1 and 3.2, which state that provided the samples $\theta_j^i$ are distributed according to the posterior, the inequalities will hold. It therefore remains only to show that the sample-then-optimize procedure yields samples from the posterior. The proof of this result can be found in Lemma 3.8 of Osband et al. [36], who show that the optimum for the gradient descent procedure described in Algorithm 1 does indeed correspond to the posterior distribution for each subset $\mathcal{D}_{<i}$.

Finally, it is straightforward to express the lower bound estimator $\hat{\mathcal{L}}$ as the sum of regression losses. We obtain this result by showing that the inequality holds for each term $\log P(\mathcal{D}_i|\theta_i)$ in the summation.

$$\log P(\mathcal{D}_i|\theta) = \log\left[\exp\left(-\frac{(\theta^\top x_i - y_i)^2}{2\sigma^2}\right)\frac{1}{\sqrt{2\pi}\sigma}\right] \qquad (23)$$

$$= -\frac{(\theta^\top x_i - y_i)^2}{2\sigma^2} - \frac{1}{2}\log(2\pi\sigma^2) \qquad (24)$$

$$= c_1 \ell_2(\mathcal{D}_i, \theta) + c_2 \qquad (25)$$

We note that in practice, the solutions found by gradient descent for finite step size and finite number of steps will not necessarily correspond to the exact local optimum. However, it is straightforward to bound the error obtained from this approximate sampling in terms of the distance of $\theta$ from the optimum $\theta^*$. Denoting the difference $|\theta - \theta^*|$ by $\delta$, we get

$$|\log P(\mathcal{D}_i|\theta^*) - \log P(\mathcal{D}_i|\theta)| = \left|\frac{((\theta^*)^\top x_i - y_i)^2}{2\sigma^2} - \frac{((\theta)^\top x_i - y_i)^2}{2\sigma^2}\right| \qquad (26)$$

$$\leq \frac{1}{2\sigma^2}|(\theta^*)^\top x_i - \theta^\top x_i|^2 \qquad (27)$$

$$\leq |((\theta^*)^\top x_i)^2 - (\theta^\top x_i)^2| + |2y||\theta^\top x - (\theta^*)^\top x| \qquad (28)$$

$$\leq |(\theta^* - \theta)^\top x + 2((\theta^*)^\top x)((\theta^* - \theta)^\top x)| + |2y||\theta^\top x - (\theta^*)^\top x| \qquad (29)$$

$$\leq |\theta^* - \theta||x| + 2|\theta^* x||\theta^* - \theta||x| + |2y||x||\theta - \theta^*| \qquad (30)$$

and so the error in the estimate of $\log P(\mathcal{D}|\theta)$ will be proportional to the distance $|\theta - \theta^*|$ induced by the approximate optimization procedure. $\qquad\square$

**Corollary 3.4.** Let $\mathcal{D}$ be a dataset indexed by our standard notation. Let $f_0$ be sampled from an infinitely wide neural network architecture $\mathcal{F}$ under some initialization distribution, and let $f_\infty^i$ be the limiting solution under the training dynamics defined by He et al. [18] applied to the initialization $f_0$ and using data $\mathcal{D}_{<i}$. Let $K_\infty$ denote the neural tangent kernel for $\mathcal{F}$, and $\mathcal{M} = GP(0, K_\infty)$ the induced Gaussian Process. Then $f_\infty^i \sim P(f|\mathcal{D}_{<i}, \mathcal{M})$, and in the limit of infinite training time, the iterative sample-then-optimize procedure yields an unbiased estimate of $\mathcal{L}(\mathcal{D}|\mathcal{M})$. Letting $\ell_2$ denote the scaled squared $\ell_2$ regression loss and $c$ be a constant, we obtain as a direct corollary of Theorem 3.3

$$P(\mathcal{D}) \geq \mathbb{E}_{f_\infty^i \sim P(\cdot|\mathcal{D}_{<i})}[\log P(\mathcal{D}_i|\theta_i)] = \mathbb{E}\sum_{i=1}^n -\ell_2(\mathcal{D}_i, f_i) + c = \mathcal{L}(\mathcal{D}) . \qquad (9)$$

*Proof.* Follows immediately from the results of He et al. [18] stating that the the limiting distribution of $f_\infty^k$ is precisely $P(f|\mathcal{D}_{\leq k}^n, \mathcal{M})$. We therefore obtain the same result as for Theorem 3.3, plugging in the kernel gradient descent procedure on $f$ for the parameter-space gradient descent procedure on $\theta$. $\square$

The following Lemma will be useful in order to prove Proposition 3.5. Intuitively, this result states that in a linear regression problem in which each feature $\phi_i$ is 'normalized' (the dot product $\langle \phi_i, y \rangle = \langle \phi_j, y \rangle = \alpha$ for some $\alpha$ and all $i, j$) and 'independent' (i.e. $\langle \Pi_{y^\perp} \phi_i, \Pi_{y^\perp} \phi_j \rangle = 0$), then the optimal linear regression solution assigns highest weight to the feature which obtains the least error in predicting $y$ on its own.

**Lemma A.1.** Let $y \in \mathbb{R}^n$, and $\Phi \in \mathbb{R}^{d \times d}$ be a design matrix such that $\Phi[:,j] = \alpha y + \epsilon_j \forall j$ for some fixed $\alpha \geq 0$, with $\epsilon \in y^\perp$, and $\epsilon_i^\top \epsilon_j = 0$ for all $i \neq j$. Let $w^*$ be the solution to the least squares regression problem on $\Phi$ and $y$. Then

$$\min_i w_i = \min_i \|f_i(x) - y\|^2 = \max_i \mathcal{L}(\mathcal{M}_i) \qquad (31)$$

*Proof.* We express the minimization problem as follows. We let $\phi(x) = (f_1(x), \ldots, f_k(x))$, where $f_i(x) = \alpha y + \epsilon_i$, with $\epsilon_i \perp \epsilon_j$. We denote by $\mathbb{1}$ the vector containing all ones (of length $k$). We observe that we can decompose the design matrix $\Phi$ into one component whose columns are parallel to $y$, denoted $\Phi_y$, and one component whose columns are orthogonal to $y$, denoted $\Phi_\perp$. Let $\sigma_i^2 = \|\epsilon_i\|^2$. By assumption, $\Phi_y = \alpha y \mathbb{1}^\top$, and $\Phi_\perp^\top \Phi_\perp = \text{diag}(\sigma_1^2, \ldots, \sigma_n^2) = \Sigma$. We then observe the following decomposition of the squared error loss of a weight vector $w$, denoted $\ell(w)$.

$$\begin{aligned}
\ell(w) &= \|\Phi w - y\|^2 = (\Phi w - y)^\top (\Phi w - y) \\
&= ((\Phi_y + \Phi_\perp)w - y)^\top ((\Phi_y + \Phi_\perp)w - y) \\
&= (\Phi_y w - y)^\top (\Phi_y w - y) + w^\top \Phi_\perp^\top \Phi_\perp w \\
&= \|y\|^2 \|1 - \alpha \mathbb{1}^\top w\|^2 + \sum \sigma_i^2 w_i
\end{aligned}$$

In particular, the loss decomposes into a term which depends on the sum of the $w_i$, and another term which will depend on the norm of the component of each model's predictions orthogonal to the targets $y$.

As this is a quadratic optimization problem, it is clear that an optimal $w$ exists, and so $w^\top \mathbb{1}$ will take some finite value, say $\beta$. We will show that for any fixed $\beta$, the solution to the minimization problem

$$\min_w \sum w_i \sigma_i^2 : w^\top \mathbb{1} = \beta \qquad (32)$$

is such that the argmax over $i$ of $w_i$ is equal to that of the minimum variance. This follows by applying the method of Lagrange multipliers to obtain that the optimal $w$ satisfies

$$w_i^* = \frac{\alpha}{\sum \sigma_i^{-2}} \frac{1}{\sigma_i^2}. \qquad (33)$$

In particular, $w_i^*$ is inversely proportional to the variance of $f_i$, and so is maximized for $i = \arg\min_i \mathbb{E}\|f_i(x) - y\|^2$.

$\square$

**Proposition 3.5.** Let $\mathcal{M}_1, \ldots, \mathcal{M}_k$ be Bayesian linear regression models with fixed noise variance $\sigma_N^2$ and Gaussian likelihoods. Let $\Phi$ be a (random) matrix of posterior prediction samples, of the form $\Phi[i,j] = \hat{y}_i^j \sim P(y_j|\mathcal{D}_{<j}, x_j, \mathcal{M}_i)$. Suppose the following two conditions on the columns of $\Phi$ are satisfied: $\mathbb{E}\langle \Phi[:,i], y \rangle = \mathbb{E}\langle \Phi[:,j], y \rangle$ for all $i, j$, and $\mathbb{E}\langle \Pi_{y^\perp} \phi_i, \Pi_{y^\perp} \phi_j \rangle = 0$. Let $w^*$ denote the least-squares solution to the regression problem $\min_w \mathbb{E}_\Phi \|\Phi w - y\|^2$. Then the following holds

$$\arg\max_i w_i^* = \arg\max_i \mathcal{L}(\mathcal{D}|\mathcal{M}_i) \qquad \forall w^* = \arg\min_w \mathbb{E}\|\Phi w - y\|^2. \qquad (11)$$

*Proof.* We first clarify the independence assumptions as they pertain to the assumptions of the previous lemma: writing $\Phi[:, i]$ as $f_i(x) + \zeta_i = \alpha y + \epsilon_i + \zeta_i$ with $\zeta_i \sim \mathcal{N}(0, \Sigma_i)$ corresponding to the noise from the posterior distribution and $f_i$ its mean, the first independence assumption is equivalent to the requirement that $f_i = \alpha' y + \epsilon_i$ with $\epsilon_i \perp y$ for all $i$. The second independence assumption is an intuitive expression of the constraint that $\epsilon_i \perp \epsilon_j$ in the linear-algebraic sense of independence, and that $\zeta_i^j$ is sampled independently (in the probabilistic sense) for all $i$ and $j$.

We note that our lower bound for each model in the linear regression setting is equal to $\mathbb{E} \sum_{i=1}^N \|f_k(x_i) + \zeta_i - y_i\|^2 + c$ where $c$ is a fixed normalizing constant. By the previous Lemma, we know that the linear regression solution $w^*$ based on the posterior means satisfies, $\max_i w_i^* = \max_i \mathcal{L}(\mathcal{M}_i)$. It is then straightforward to extend this result to the noisy setting.

$$\mathbb{E}[\|\Phi w - y\|^2] = \mathbb{E}[\|(\Phi_y + \Phi_\perp + \zeta)w - y\|^2] \tag{34}$$

$$= \mathbb{E}[((\Phi_y + \Phi_\perp + \zeta)w - y)^\top ((\Phi_y + \Phi_\perp + \zeta)w - y)] \tag{35}$$

$$= \|\Phi_y w - y\|^2 + w^\top \Phi_\perp^\top \Phi_\perp w + \mathbb{E}[w^\top \zeta^\top \zeta w] \tag{36}$$

$$= (w^\top \mathbb{1} - \alpha)^2 \|y\|^2 + w^\top \Phi_\perp^\top \Phi_\perp w + \mathbb{E}[w^\top \zeta^\top \zeta w] \tag{37}$$

$$= (w^\top \mathbb{1} - \alpha)^2 \|y\|^2 + \sum w_i^2 (\|\Phi_\perp[:, i]\|^2 + \|\zeta_i\|^2) \tag{38}$$

We again note via the same reasoning as in the previous Lemma that the model with the greatest lower bound will be the one which minimizes $\|\Phi_\perp[:, i]\|^2 + \|\zeta_i\|^2$, and that the weight given to index $i$ will be inversely proportional to this term.

It only remains to show that for each model $i$, the model which maximizes $\mathcal{L}(M_i)$ will also minimize $\|\Phi_\perp[:, i]\|^2 + \|\zeta_i\|^2$. This follows precisely from the Gaussian likelihood assumption. As we showed previously

$$\mathcal{L}(\mathcal{D}|\mathcal{M}_i) = \mathbb{E}[\sum \log P(y_i|\mathcal{D}_{<i})] \propto - \sum \mathbb{E}[\ell_2(y_i - \hat{y}_i] \tag{39}$$

$$= [\|y - \mu\|^2 + \mathbb{E}[\|\hat{y} - \mu\|^2] \tag{40}$$

$$= \alpha \|y\|^2 + \|\Phi_\perp[:, i]\|^2 + \mathbb{E}[\|\zeta_i\|^2] \tag{41}$$

and so finding the model $\mathcal{M}_i$ which maximizes $\mathcal{L}(\mathcal{D}, \mathcal{M}_i)$ is equivalent to picking the maximal index $i$ of $w^*$ which optimizes the expected loss of the least squares regression problem. $\square$

Figure 5: Relative rankings given by optimize-then-prune, ML, and estimated $\mathcal{L}(\mathcal{D})$. Left: feature selection. Middle: prior variance selection. Right: RFF frequency selection. Rankings are consistent with what our theoretical results predict. Results are averaged over 5 runs.

# B Experiments

## B.1 Experimental details: Model Selection using Trajectory Statistics

We consider 3 model selection settings in which to evaluate the practical performance of our estimators. In **prior variance selection** we evaluate a set of BLR models on a synthetic linear regression data set. Each model $\mathcal{M}_i$ has a prior distribution over the $d$ parameters of the form $w \sim \mathcal{N}(0, \sigma_i^2 I_d)$ for some $\sigma_i^2$, and the goal is to select the optimal prior variance (in other words, the optimal regularization coefficient). We additionally evaluate an analogous initialization variance selection method on an NTK network trained on a toy regression dataset. In **frequency (lengthscale) selection** we use as input a subset of the handwritten digits dataset MNIST given by all inputs labeled with a 0 or a 1. We compute random Fourier features (RFF) of the input to obtain the features for a Bayesian linear regression model, and perform model selection over the frequency of the features (full details on this in the appendix). This is equivalent to obtaining the lengthscale of an approximate radial basis function kernel. In **feature dimension selection**, we use a synthetic dataset [46] of the form $(\mathbf{X}, \mathbf{y})$, where $x_i = (y_i + \epsilon_1, y_i + \ldots, y_i + \epsilon_{15}, \epsilon_{16}, \ldots, \epsilon_{30})$. We then consider a set of models $\{\mathcal{M}_k\}$ with feature embeddings $\phi_k(x_i) = x_i[1, \ldots, k]$. The optimal model in this setting is the one which uses exactly the set of 'informative' features $x[1, \ldots, 15]$.

The synthetic data simulation used in this experiment is identical to that used in [46]. Below, we provide the details.

Let $k$ be the number of informative features and $d$ the total number of features. We generate a datapoint $\mathcal{D}_i = \{x_i, y_i\}$ as follows:

1. Sample $y_i$: $y_i \sim U([0, 1])$
2. Sample $k$ informative features: $x_{i,j} \sim N(y_i, \sigma_0) \quad \forall j \in 1, \ldots k$
3. Sample $\max(d - k, 0)$ noise features: $x_{i,k+j} \sim N(0, \sigma_1) \quad \forall j \in 1, \ldots d - k$
4. Concatenate the features: $X_i = [x_{i,1}, \ldots x_{i,d}]$

We set $\sigma_0 = \sigma_1 = 1$, $k = 15$, $n = 30$, and let $d$ vary from 5 to $n$. We then run our estimators on the Bayesian linear regression problem for each feature dimension, and find that all estimators agree on the optimal number of features, $k$.

To compute the random fourier features used for MNIST classification, we vectorize the MNIST input images and follow the procedure outlined by [37] (Algorithm 1) to produce RFF features, which are then used for standard Bayesian linear regression against the binarized labels. The frequency parameter (which can also be interpreted as a transformation of the lengthscale of the RBF kernel approximated by the RFF model) is the parameter of interest for model selection.

## B.2 Experimental details: Bayesian model comparison

Here we provide further detail of the experiment in Section 4.2.1. The goal of the experiment is to determine whether the connection between sum-over-training losses (SOTL) and model evidence observed in the linear regression setting extends to DNNs. In particular, the two sub-questions are:

1. Do models with a lower SOTL generalize better?

2. Are these models favoured by SGD?

To answer these questions, we train a linear combination of NNs. We can answer subquestion [1] by plotting the correlation between SOTL and test performance of an individual model. Further, we address subquestion [2] by considering the correlation between test loss and linear weights assigned to each model.

Below we explain the set-up of the linear combination in more detail. We train a variety of deep neural networks along with a linear 'ensemble' layer that performs a linear transformation of the concatenated logit outputs[3] of the classification models. Let $h_m(x_i)$ be logit output of model $m$ for input $x_i$, $\ell(y_i, h_i)$ be the loss for point $i$ (where $h_i$ is a logit) and $w_{m,t}$ be the weight corresponding to model $m$ at time step $t$.

We consider two training strategies: we first train models individually using the cross-entropy loss between each model's prediction and the true label, only cross-entropy loss of the final ensemble prediction to train the linear weights. Mathematically, we update the models using the gradients

$$\frac{\delta}{\delta \theta_m} \ell(y_i, h_m(x_i)), \tag{42}$$

and the 'ensemble' weights using

$$\frac{\delta}{\delta w_m} \ell(y_i, \sum_m w_m h_m(x_i)). \tag{43}$$

We refer to this training scheme as *Parallel Training* as the models are trained in parallel. We also consider the setting in which the models are trained using the cross entropy loss from the ensemble prediction backpropagated through the linear ensemble layer, i.e. the model parameters are now updated using:

$$\frac{\delta}{\delta \theta_m} \ell(y_i, \sum_m w_m h_m(x_i)). \tag{44}$$

We refer to this scheme as the *Concurrent Training*.

We train a variety of different MLPs (with varying layers,and nodes) and convolutional neural networks (with varying layers, nodes and kernels) on FashionMNIST using SGD until convergence.

### B.3 Experimental Details: SGD upweights submodels that perform well

Below we provide further details of the experiment in Section 4.2.2. The goal of the experiment is to determine whether SGD upweights sub-models that fit the data better.

We train a MLP network (with units $200, 200, 10$) on FashionMMIST using SGD until convergence. After training is completed, for every class of $y$, we rank all nodes in the penultimate layer by the norm of their absolute weight (in the final dense layer). We group the points into submodels according to their ranking – the $k$ nodes with the highest weights are grouped together, next the $k+1, \ldots 2k$ ranked nodes are grouped, etc. We set $k = 10$.

We determine the performance of a submodels by training a simple logistic classifier to predict the class of an input, based on the output of the submodel. To measure the performance of the classifier, we use the cross-entropy loss. To capture the equivalent notion of the AUC, we estimate the performance of the sub-models throughout training, and sum over the estimated cross-entropy losses.

Below, we show additional plots for the *parallel* and *concurrent* training schemes. The results are the same to those presented in the main text, and we observe [1] a negative correlation between test performance and ensemble weights and [2] a strong correlation between SOTL and average test cross-entropy.

Figure 6: **Linear combinations of DNNs on FashionMNIST.** Left: ensemble weights versus the test loss for parallel training; we observe a negative correlation. Middle: SOTL (standardized by the number of training samples) versus test loss for concurrent and concurrent training. We observe a strong correlation indicating that the SOTL generalizes well. Right: training curves for the different models in concurrent training schemes. All results are averaged over 10 runs, and standard deviations are shown by the shaded regions around each observation. The model parameters, given in the parentheses, are the number of layers ($l$), nodes per layer ($n$) and kernel size ($k$), respectively.

However, similarly to the linear setting, the difference in assigned weights is magnified in the concurrent training scheme. Here we find that in the concurrent training scheme, the ensemble focuses on training the CNNs (as can be seen from the training curve in Figure 3 in the main text). This is likely because CNNs are able to learn more easily, leading to larger weights earlier on.

Figure 7: Weight assigned to subnetwork by SGD in a deep neural network (x-axis) versus the subnetwork performance (estimated by the sum of cross-entropy, on the y-axis) for different FashionMNIST classes. The light blue ovals denote depict 95% confidence intervals, estimated over 10 seeds (i.e. $2\sigma$ for both the weight and SOTL). The orange line depicts the general trend.

Figure 8: Weight assigned to subnetwork by SGD in a deep neural network (x-axis) versus the subnetwork performance (estimated by the sum of cross-entropy, on the y-axis) for different CIFAR-10 classes. The light blue ovals denote depict 95% confidence intervals, estimated over 10 seeds (i.e. $2\sigma$ for both the weight and SOTL). The orange line depicts the general trend.

Above, we show additional plots to those shown in Figure 4, Section 4.2.2. Figure 7 shows the results for the all FashionMNIST classes, and Figure 8 shows the results for experiment on CIFAR-10. From both, we see that SGD assigns higher weights to subnetworks that perform better.

## Footnotes

[3]These are pre-softmax outputs. To obtain the predicted probability of a class, they are fed through a softmax function.