[Reviews · NeurIPS 2020]

Review 1

Summary and Contributions: The paper relates training speed and the model evidence from a Bayesian perspective. In particular, a measurement of training speed is a lower bound on model evidence. This uncovers a Bayesian model selection method via sample-then-optimize fashion. Computational results are provided in toy dataset and MNIST-fashion.

Strengths: 1. The paper is well written, the mathematical presentation is easy to understand. 2. The connection between training speed and the model evidence is interesting, and as far as I know the main idea is original. Equation (2) is simple yet insightful, I like the idea that the log evidence is equivalent to the area under a training curve. 3. Theoretical analyses for linear models are provided. Although the results do not directly generalize for non-linear models, the theory is also applicable empirically. 4. The proposed method is simple yet effective, which only need an additional linear layer over the models.

Weaknesses: 1. The "training speed" analyzed in this paper could be very different from the previous works. Equation (2) assumes the update follows a Bayesian updating procedure. However, stochastic optimization are adopted for most of thee gradient-based optimization algorithms. 2. The experiments are conducted only in toy dataset such as MNIST-fashion. I am wondering does the proposed model selection approach scale beyond MNIST-like dataset? It would also be interesting to compare the proposed approach with other neural network pruning methods. 3. There should be more baselines in the experiments sections. For instance, for feature dimension selection, the proposed approach should at least compare with LASSO. More simple baselines could provide a better overview.

Correctness: The main message is reasonable and the experiments are sound.

Clarity: The paper is well-written.

Relation to Prior Work: Most of the works related to disentanglement are included. Clear discussions are provided.

Reproducibility: Yes

Additional Feedback: 1. The L(D) defined in equation (2) is a lower bound on the log evidence of D. However, in section 3.2.1 L is used for describing the evidence of a model. Does the L ini Thm 1 the same L in (2)? 2. Adding a linear network for selecting model (proposition 1) sound fairly intuitive. Based on the proposition, why not directly choose the model that achieve the lowest empirical error? After reviewing author's response, most of my concern has been addressed or discussed. I still look forward to more comparisons or discussions with non-Bayesian approach. Nevertheless, this is a well-written and interesting paper and I will not change my score.


Review 2

Summary and Contributions: The authors attempt to reach an understanding of (1) what could explain the relationship between training speed and model generalization performance, and (2) why SGD leads to good generalization even in overparametrized regimes where it could easily overfit. I will both summarize the paper and authors' contributions under these two questions. Contribution 1: The authors observe that the alternative expression for marginal likelihood as the product of consecutive posterior predictive likelihoods, when negated, can be interpreted as corresponding to iterative losses in training with SGD. Building on a theoretical result from the literature that shows initial parameters sampled from the prior and trained under different models lead to a set of posterior parameter samples, the authors show that cumulative SGD losses can be interpreted as an estimator for the marginal likelihood or a lower bound of it. This serves as a potential explanation for the previously observed relationship between training speed and generalization. Contribution 2: The authors then show that SGD assigns higher weights to component linear models that have lower cumulative training losses. As the cumulative training losses were argued to be approximations for the marginal likelihoods of a model, SGD can now be interpreted to conduct an approximation of marginal likelihood based model selection. The authors propose this observation as another potential explanation for the generalization properties of SGD. After obtaining theoretically backed results in linear models, the authors examine whether a similar observation pertains to neural networks trained with SGD as well: though they establish that SGD chooses subnetworks with low sum of training errors and high generalization, relation to marginal likelihood in this case is more uncertain.

Strengths: I believe that this article contributes to three timely, related issues: the relationship between training speed and generalization, explaining SGD's generalization properties, and probabilistic interpretations of neural network behavior. I believe that it is sufficiently relevant for the NeurIPS community. Throughout the paper, the authors' arguments retain their cohesiveness, and the paper ends with having made interesting and relevant observations. On the other hand, there are certain conceptual and organizational issues that mar this process (see below). As long as these issues are addressed or appropriately discussed, I believe the authors' observations and conclusions are worthy of attention. Lastly, based on the authors' last experiment, I believe that the idea of SGD as inducing a cascade of marginal likelihood based model selections for all subnetworks is an interesting idea. However first, establishing a more concrete relationship between the marginal likelihood-based model selection arguments in the linear models and the results in the deep learning experiments is necessary.

Weaknesses: At Eq. 5, the authors introduce two sampling based estimators of the lower bound (LB). I am not sure why the authors introduced both as an estimator for the LB: The second estimator is an unbiased estimator of the (log) marginal likelihood (ML). Though it could technically be considered a biased estimator of LB, I do not see why it should be introduced as such, since it is the unbiased estimator of the exact value the authors are hoping to approximate. Actually, in the following sentence the authors write that the second estimator's bias decreases as |J| is increased, which is very much expected, if not almost trivial considering the point above. Another point is that when |J| = 1 the two estimators are algebraically the same, therefore the first one also becomes (a noisy) unbiased estimator of ML. This is another point that might deserve mentioning, even if in Appendix. It is also confusing that the authors, in addition to ignoring the fundamental difference between the two, do not name them differently, and keep referring to both of them as the estimator of the lower bound, usually without qualification. Importantly, moving forward, the authors are not explicit about which estimator they are going to use at default (as they call both estimators lower bound estimators). The choice of estimator is not a trivial one given the difference mentioned above. I deduced that since the second figure was introduced to show the superiority of the second estimator (of ML), the rest must have used the first estimator; this should have been made clear by the authors. Since the authors moved forward with the first estimator, I assumed that this was due them being able to obtain theoretical guarantees only for that one. But this leaves the question about their experiments in Figure 2 which they presented without any reference to lack of theoretical results. Lastly, after having demonstrated the results for the linear models in Section 4.1, I found that the transition to the neural networks remains a bit confusing. Although their results regarding sum of training loss (SOTL) and generalization is interesting regardless, in Section 4.2.1 the results are presented as SGD upweighing models with low SOTL - high generalization, but in Section 4.2.2 similar results are presented as implying approximate Bayesian model selection. I think sections on experiments with neural networks could be expressed more unequivocally with regard to how much the results support the marginal likelihood-based model selection hypothesis, or whether they should more safely be observed only as a relationship between SOTL and generalization.

Correctness: I voiced my potential problems with the authors argumentation in the previous section, and I think that authors' empirical evaluations are fit for their purpose. Here I will voice a few miscallenaous issues: - I was not able to ascertain how the result of Theorem 2 is used in the text, I'd be happy if the authors could clarify. - (Pg. 3) Section 3.1: I assume the authors must have meant taking the negative of the posterior predictive likelihood as the loss. - (Pg. 3) Section 3.1: log expected likelihood under the current posterior (i.e. P(theta|D_i)) -> did the authors mean P(theta|D_<i)? A similar issue is present at section 3.2 right above Eq. 5. (P(theta|D_i-1)) - I imagine there might be a problem with Figure 1, it describes two results but depicts three graphs.

Clarity: I think the paper in general is not hard to follow, except for the problems mentioned above and below. The supplementary material seem to have changed sections/letters after the main document was finalized. So the references to the appendices seem to be off. In the experiments, I found the fact that the authors call the exact sampling estimator of the lower bound "exact ELBO" because the lower bound does have a (theoretical) exact value; so something in the line of "exact sampling ELBO" could be more appropriate. Also the authors refer to the same value as "ELBO" in the Fig. 1 which is further confusing. The experiments depict the value "log evidence". Is this the analytic computation of the value? The authors use the abbreviation ELBO for their evidence lower bound. Though technically correct (as in being an Evidence Lower BOund), I would consider using a different abbreviation given the association of the term with the variational lower bound in variational inference, or maybe discuss the relationship between the two. Some other small points: - (Pg. 3) The authors have introduced \mathcal{D}^n for the data set but then go on to use \mathcal{D} starting Eq. 2. - (Pg. 4) The authors introduce the notation \mathcal{L}(D) but then switch to notation \mathcal{L}(M_1) - (Pg. 5) Proposition 1: mean zero -> zero mean - (Pg. 8) Section 4.2.2: "are by gradient descent" -- missing word?

Relation to Prior Work: I found the literature review to be satisfactory and authors are articulate in expressing how their work fits in the literature. One potential point of improvement would be to hear authors' opinion on the discrepancy between the "cold posterior" results mentioned above and the rest of the literature they cited which favor Bayesian model selection approaches in generalization. Also, authors might want to point out the potential parallels between their results in Section 4.2.2 and those in Frankle and Carbin (2019) that they cited [7].

Reproducibility: Yes

Additional Feedback: Additional comments after author feedback ================================ I thank the authors for their detailed feedback, we seem to agree on the majority of my points. Some of my comments were understandably unaddressed due to space limitations, I believe they will be taken into consideration in further editing of the paper. However, I still do not think we agree on a point of contention, and I voice my concerns below. That being said, I believe that (potential) problems to arise from that point can be resolved, and paper's contributions warrant its acceptance. I hope that the comments below will be helpful in preparing the authors' work for dissemination. "We affirm that both estimators compute unbiased estimates of..." - I stand corrected, the authors are right that the 2nd estimator at (5) is indeed a biased estimator of log-ML with finite |J|. Its exponential is the unbiased estimator of ML, so the correct property for this discussion would be consistency. The 2nd estimator at (5) is a consistent estimator of log-ML*. Thus, it is an inconsistent estimator of the lower bound at (3), with the bias P(D|M) - L(D|M) at the infinite sample limit. Therefore, I am still not clear on why the 2nd estimator should be presented as an estimator of the specific lower bound at (3). Figure 2 partly reinforces the confusion above by evaluating the bias of this estimator against log-ML. More importantly, in its current form, I suspect that the proof of Theorem 1 seems to apply only to the 1st estimator at (5), as the equation below line 125 would not apply to the 2nd estimator. We would need a constant d(J, D, M_i) <= P(D|M_i) - L(D|M_i) to account for the expected difference between the two estimators due to Jensen's equality, which might invalidate the bounding by C (and it might disrupt the ordering between the lower bounds of the models since it is model dependent). I believe that the authors should either discuss how Theorem 1 still holds under the 2nd estimator or use the 1st estimator for the experiments. "We will clarify the number..." & "Because both discussed estimators..." & "the results of Figure 2 are in fact consistent with our theory" - Apart from the discussion above, the main point of my ensuing comments was that I did not understand which estimator in (5) was used in which experiments. From the authors' feedback I infer that they always used the second estimator (is this correct?). It would be great if the authors were explicit about this. * Log-MC estimators for each term i in the log-ML sum are consistent due to continuous mapping theorem as they are each log transforms of consistent estimators. Sum of consistent estimators is consistent. ================================ Interestingly, in the extensive article by Jiang et al. 2020 (Fantastic Generalization Measures), the training speed results seem to be interpreted in a confusing fashion. While the authors find a negative correlation between number of iteration steps and generalization "the less steps to 0.1 CE error, the higher generalization"; they interpreted it as "the slower it the training, the higher generalization". My interpretation of this was similar to the current paper's authors': I accepted their numerical, former results. But since these results are central to current authors' work, they might want to ascertain that this indeed was the case. Also, the authors might be interested in reading: https://arxiv.org/abs/1905.08737. I also suspect this (post-NeurIPS) article to be relevant to their pursuit: http://arxiv.org/abs/2006.15191.


Review 3

Summary and Contributions: The paper studies the connection between training speed and Bayesian model evidence. The paper presents an intersting connection between model training speed and marginal likelihood. Moreover, the paper shows that SGD performs model selection based on the marginal likelihood.

Strengths: The claims presented in the paper are supported both empirically and theoretically. I think it is important for the community to see these results.

Weaknesses: Figures 1,2, and 3 needs clearer explanation about the experiment setup.

Correctness: Yes, to the best of my knowledge.

Clarity: I found the beginning of the paper and the motication easy to follow. However, for the experimental setup, I think it needs clearer explanation.

Relation to Prior Work: The paper tries to explain the observations in the literature of SGD being biased to simple solutions that generalize better.

Reproducibility: Yes

Additional Feedback:

[Author Response · NeurIPS 2020]

We thank the reviewers for their helpful feedback. We were glad to see that all reviewers understood and remarked on both our theoretical contributions showing an equivalence between a notion of training speed and the Bayesian model evidence in linear models, and our empirical results confirming 1) the theory in the linear setting and 2) showing similar mechanisms at play in neural networks. In particular, **R1**: The connection between training speed and the model evidence is interesting, and [...] the main idea is original; and **R2**: I believe the authors' observations and conclusions are worthy of attention. We now address some concerns.

**R1**: 'The experiments are conducted only in toy dataset.' We have replicated the DNN experiments (S4.2) on the CIFAR-10 dataset and observe similar results as for FashionMNIST; as can be seen in Figure 1 below.

**R1**: 'Equation (2) assumes the update follows a Bayesian updating procedure. However [...]' We leverage the work of Matthews et al. [citation 15 in the paper] demonstrating that in the linear setting, gradient descent is approximately equivalent to Bayesian posterior updating. Our experiments on linear models all follow this procedure, consistent with our theory. Extending our results to multi-epoch SGD and the nonlinear setting is an exciting avenue for future work.

**R2**: 'The second estimator is an unbiased estimator of the (log) marginal likelihood (ML)'. We thank

Figure 1: Replication of Figure 5 on CIFAR-10.

the reviewer for bringing this potential source of confusion to our attention. We affirm that *both* estimators compute unbiased estimates of members of a family of lower bounds on the marginal likelihood (defined with respect to the number of samples $J$) and so will produce biased estimates of the marginal likelihood. We can derive this result using Jensen's inequality as follows: $\mathbb{E}_{\theta_j^i \sim P(\theta|\mathcal{D}_{<i})}[\sum_{i=1}^n \log \frac{1}{J} \sum_{j=1}^J P(\mathcal{D}_i|\theta_j^i) \leq \sum_{i=1}^n \log \mathbb{E}_{\theta_j \sim P(\theta|\mathcal{D}_{<i})} \frac{1}{J}[\sum_{j=1}^J P(\mathcal{D}_i|\theta_j^i)] = \sum_{i=1}^n \log P(\mathcal{D}_i|\mathcal{D}_{<i})$. I deduced [...] this should have been made clear by the authors. We will clarify the number of samples used to estimate $\log P(\mathcal{D}_i|\mathcal{D}_{<i})$ in our revisions; thanks for highlighting this. [...] this leaves the question about their experiments in Figure 2 which they presented without any reference to lack of theoretical results. Because both discussed estimators are for lower bounds to the log ML (and we know that Jensen's inequality becomes tighter with reduced variance), the results of Figure 2 are in fact consistent with our theory, which is developed in section 3.2.1.

**R2**: 'interpretation of [Jiang et al.].' We agree that the wording of Jiang et al. is ambiguous; we have verified the numerical results from the paper and are reasonably confident that our interpretation is correct.

'I was not able to ascertain how the result of Theorem 2 is used in the text, I'd be happy if the authors could clarify.' Theorem 2 is the noiseless analogue of Theorem 1, and we include it to demonstrate that it is still possible to obtain an estimate of the model evidence based on training statistics for linear models in the noiseless setting. The reviewer is correct that we do not follow up on this result empirically; this is to save space for experiments that more clearly link training speed, generalization, and the marginal likelihood.

**R1**: 'There should be more baselines in the experiments sections [...] LASSO [...]' We agree that a comparison between Bayesian model selection and other model selection baselines is an important line of work; unfortunately, a thorough empirical analysis of the generalization performance of Bayesian model selection compared to non-Bayesian baselines was outside the scope of this work. Because we show that our method obtains a similar ranking over models as Bayesian evidence maximization, we can extrapolate that it will exhibit similar strengths and weaknesses as exact marginal likelihood maximization when compared against other baselines such as LASSO.

**R2:** 'I found that the transition to the neural networks remains a bit confusing.' We agree that the analysis of neural networks is quite distinct from our results in the linear setting, and we believe that bridging from linear models to neural networks via a discussion of infinitely wide neural networks (the neural tangent kernel regime) will clarify this transition. Our sample-then-optimize lower bound estimator for linear models can be applied in a straightforward way to gradient descent on infinitely wide networks trained using the procedure outlined in [1] (posted after our submission to NeurIPS). We have updated the paper to discuss this corollary of our results as a way to better motivate and provide context for the neural networks results.

'how much the results support the marginal likelihood-based model selection hypothesis, or whether they should more safely be observed only as a relationship between SOTL and generalization.' We thank the reviewer for highlighting this. Our empirical results for DNNs are presented as evidence that the mechanism that we observe in linear models seems to be at work in deeper models as well, as a way to motivate further exploration of the ideas presented here. We will clarify this aspect in our paper.

**R3**: 'Figures 1,2, and 3 needs clearer explanation about the experiment setup.' We thank the reviewer for bringing up this source of confusion and will clarify this in our edits: due to the rebuttal's page limit, we defer to the discussion of experimental setup in the appendix if more details are needed.

[1]Bobby He, Balaji Lakshminarayanan, and Yee Whye Teh. Bayesian deep ensembles via the neural tangent kernel.

[Meta-Review · NeurIPS 2020]

The work considers SGD training for Bayesian linear models and illustrate a connection between training speed and generalization and why SGD tends to select simpler models. In particular, the work illustrates that a particular type of posterior sampling from gradient descent yields same model rankings as that based on the true posterior under suitable assumptions. Experiments on deep nets are also presented. The reviewers liked the work overall, but felt that some aspects of the exposition were unclear, the transition and implications for deep nets is not quite convincing especially since there is now better understanding of both optimization and generalization in deep nets, and baseline comparisons (e.g., sgld, L2 regularization, dropout, etc.) would strengthen the work.